# The Ins and Outs of Homeodomain-Leucine Zipper/Hormone Networks in the Regulation of Plant Development

**DOI:** 10.3390/ijms25115657

**Published:** 2024-05-23

**Authors:** Giovanna Sessa, Monica Carabelli, Massimiliano Sassi

**Affiliations:** Istituto di Biologia e Patologia Molecolari, Consiglio Nazionale delle Ricerche, 00185 Rome, Italy; giovanna.sessa@cnr.it (G.S.); monica.carabelli@cnr.it (M.C.)

**Keywords:** HD-ZIP, transcription factors, plant development, phytohormones, shoot meristem, root meristem, leaf development, embryogenesis

## Abstract

The generation of complex plant architectures depends on the interactions among different molecular regulatory networks that control the growth of cells within tissues, ultimately shaping the final morphological features of each structure. The regulatory networks underlying tissue growth and overall plant shapes are composed of intricate webs of transcriptional regulators which synergize or compete to regulate the expression of downstream targets. Transcriptional regulation is intimately linked to phytohormone networks as transcription factors (TFs) might act as effectors or regulators of hormone signaling pathways, further enhancing the capacity and flexibility of molecular networks in shaping plant architectures. Here, we focus on homeodomain-leucine zipper (HD-ZIP) proteins, a class of plant-specific transcriptional regulators, and review their molecular connections with hormonal networks in different developmental contexts. We discuss how HD-ZIP proteins emerge as key regulators of hormone action in plants and further highlight the fundamental role that HD-ZIP/hormone networks play in the control of the body plan and plant growth.

## 1. Introduction

One of the fundamental questions in developmental biology is how different shapes of multicellular organisms are determined by the genetic properties of an individual. It is widely accepted that patterns of signaling molecules—often referred to as morphogens—are established in clusters of cells at a specific developmental stage and determine the identity of those cells and eventually their fate. Morphogen patterns do not only determine whether the cells are bound to proliferate or differentiate, but they also provide the spatial and temporal parameters that regulate the dynamics of tissue growth [1,2]. In plants, where cell migration is impeded by the presence of the cell wall and programmed cell death is restricted to a few tissues, the regulation of growth dynamics underlying shape emergence depends on the localized patterns of morphogen-like molecules like transcription factors and phytohormones [3,4]. Indeed, molecular regulatory networks encompassing transcriptional cascades and hormonal signaling play key roles in determining the basal body shape during embryogenesis as well as the whole architecture of adult plants. Hormone/TF networks also represent a hallmark of plant evolution as the recruitment of pre-existing signaling molecules, the generation of new ones via genome duplication and neofunctionalization, and their assembly into novel developmental genetic toolkits in a sort of “genetic bricolage” allow for the generation and further diversification of new plant architectures required for land colonization [5,6,7]. Key morpho-physiological innovations that characterize land plants, such as the presence of a root for soil anchorage, the development of a branched structure with expanded appendices for improved photosynthesis, the presence of a cuticle, and a vasculature for better resistance to desiccation, have been accompanied by the expansion and diversification of hormone signaling machinery and of families of transcriptional regulators [6,8]. HD-ZIP transcription factors play a pivotal role in the adaptation of plants on lands [9], and there is evidence that HD-ZIPs have been recruited multiple times during evolution to alter organ shapes and generate new specialized structures [10,11,12]. An increasing body of works demonstrates that HD-ZIP TFs exert their developmental role by controlling the accumulation, distribution, and signaling of multiple hormones. Here, we will review our current knowledge on HD-ZIP/hormone networks in the regulation of plant architecture and tissue growth across different developmental stages of the model plant *Arabidopsis thaliana*.

## 2. HD-ZIP Protein Structure

The *Arabidopsis thaliana* genome codes 48 HD-ZIP proteins that have been grouped into four families based on sequence homology in the HD-ZIP domain, the presence of additional conserved motifs, and specific intron and exon positions: HD-ZIP I (17 members), HD-ZIP II (10 members), HD-ZIP III (5 members), and HD-ZIP IV (16 members) [13]. From a structural standpoint, HD-ZIP proteins are characterized by the presence of a homeodomain (HD) that is closely linked to a leucine zipper (LZ) motif. The HD is crucial for recognition and interaction with DNA target sequences, whereas the LZ domain enables homodimerization, which is a prerequisite for DNA binding [14]. Binding site selection analysis and subsequent chromatin immunoprecipitation sequencing experiments have determined that HD-ZIP proteins recognize pseudo-palindromic DNA elements [15]. Interestingly, the identified cis elements recognized by HD-ZIP I, II, and III share the consensus sequence AAT(N)ATT, suggesting that members of these three different families may regulate common target genes [15]. The HD-ZIP IV target sequence is slightly different and overlaps with the L1 box TAAATG(C/T)A [16].

Besides the HD-ZIP motif, these TFs have other functional domains that characterize each family. Family I proteins are characterized by the presence of an AHA transcriptional activation domain [17]. At the N-ter, family II proteins contain an LxLxL-type EAR motif typical of negative regulators of gene expression and a CPSCE motif at the C-ter, which may serve as a redox sensor [18,19]. HD-ZIP III proteins are characterized by a START domain, which possesses lipid-binding properties [20], a START-associated domain named SAD, and a MEKHLA domain which shows sequence similarity with a known light, oxygen, or redox sensor, namely the PAS domain [21]. Similarly to family III, HD-ZIP IV proteins contain START and SAD domains but lack the MEKHLA at the C-ter [22]. Recent works have shown that START domain plays a key role in protein dimerization, DNA binding and transcriptional activity of family III and IV proteins [23,24].

## 3. Embryogenesis

Embryo morphogenesis consists of a series of major patterning events that establish the basic shape of a plant, which is defined by the establishment of (1) the apical–basal axis, (2) the radial axis, and (3) bilateral symmetry. These patterning events are controlled by the activity of several TFs through complex regulatory networks interconnected with hormonal pathways. HD-ZIP TFs are key components in such networks as they have been shown to intervene in different aspects of embryo development [25].

Members of the HD-ZIP IV family play key roles in the early stages of embryo development. HOMEODOMAIN GLABROUS (HDG) 11 and HDG12 function in the establishment of the apical–basal axis by regulating the asymmetric division of the zygote [26]. Maternally derived HDG11 and HDG12 were shown to directly regulate the expression of *WUSCHEL-RELATED HOMEOBOX 8 (WOX8)*, a key regulator of zygote asymmetry and embryo development [26,27]. WOX8—along with paralog WOX9—regulates the expression of the auxin (IAA) efflux carrier *PIN-FORMED1 (PIN1)* and, as a consequence, IAA distribution during embryogenesis [28], suggesting that HDG11/12 affects IAA-mediated patterning via WOX TFs [26]. However, HD-ZIP IV TFs could also interfere with IAA-mediated embryo development by other means. Several HD-ZIP IV TFs were shown to interact with members of the PLETHORA (PLT)/AINTEGUMENTA (ANT)/AINTEGUMENTA-LIKE(AIL) family of APETALA2/ETHYLENE RESPONSE FACTOR (AP2/ERF) TFs, including ANT and PLT4/BABY BOOM (BBM) [29], which represent key regulators of developmental responses downstream of IAA [30,31,32]. The co-suppression of HDG11, HGD12, and HDG1 expression cause somatic embryogenesis in adult tissues, analogously to what was observed for BBM overexpression [29,33]. Coherently, HDG1 and BBM oppositely regulate a subset of genes involved in IAA, cytokinin (CK), and gibberellic acid (GA) responses [29]. ARABIDOPSIS THALIANA MERISTEM LAYER 1 (ATML1) and PROTODERMAL FACTOR 2 (PDF2) were also shown to interact with BBM and ANT in vitro [29], although the functional relevance of these interactions in planta remains unknown. ATML1 and PDF2 are master regulators of protoderm formation, and despite complex regulatory mechanisms underlying the establishment and maintenance of their epidermal localization during embryogenesis [34,35,36,37,38], no interactions with specific hormone signals have been unraveled so far.

Family III HD-ZIPs are pivotal to the regulation of embryo polarity and determine the apical fate by antagonizing the function of auxin-regulated, root-pole-specific PLT/AIL TFs [39] (Figure 1). The expression of *HD-ZIP III* genes is restricted to the upper cell tier of the embryo following non-cell-autonomous regulation by microRNA 165 and 166 (miR165/6) [40,41,42,43,44]. The disruption of the HD-ZIP III expression domain alters embryo polarity: the misexpression of miR-resistant *HD-ZIP III* transcripts transform the root pole into a secondary shoot [39]. In contrast, the PLT/AIL members PLT1, PLT2, PLT3/AIL6, and PLT4/BBM are expressed in an IAA-dependent manner in the lower cell tier of the embryo and determine the root pole fate [30,31,45].

*PLT1* and *PLT2* are direct targets of TOPLESS (TPL), a transcriptional corepressor involved in auxin responses downstream of Skp1-Cullin-F-Box (SCF) TRANSPORT INHIBITOR RESISTANT1/AUXIN SINALING F-BOX (TIR1/AFB) receptor complexes [39,45,46,47]. *tpl-1* mutations lead to a shoot-to-root fate change during embryogenesis as a result of an auxin-dependent PLT misexpression in the apical pole and subsequent alteration in PIN4-dependent IAA transport [39]. Gain-of-function mutations in *HD-ZIP III* genes restore apical fate in *tpl-1* by repressing the PLT pathway in the shoot domain [39], indicating that HD-ZIP III TFs repress auxin-mediated root pole specification (Figure 1).

REVOLUTA (REV), PHABULOSA (PHB), PHAVOLUTA (PHV), and CORONA (CNA) also play redundant roles in the establishment of the bilateral symmetry and shoot apex from the heart stage onwards, as exemplified by the alterations observed in multiple loss-of-function mutant combinations ranging from single cotyledon to a complete loss of the embryonic shoot apical meristem (SAM) [40,48,49]. These phenotypes are derived from an antagonistic interaction with the KANADI (KAN) family of TFs in the control of PIN1-mediated IAA transport routes during embryogenesis [50]. In *phv phb rev* transition stage embryos, PIN1 fails to reverse its polarity towards cotyledons primordia in the presumptive shoot apex, leading to defects in bilateral symmetry; in *kan1 kan2 kan4* embryos, PIN1 is ectopically expressed in distinct foci below the presumptive cotyledons, leading to the production of ectopic leaf-like structures in the seedling hypocotyl [50]. Hextuple mutants between *kan* and *hd-zip III* loss-of-function alleles partially mitigate *hd-zip III* phenotypes, suggesting that auxin transport defects in *phv phb rev* are caused by the ectopic expression of *KAN* genes [50] (Figure 1).

It is interesting to note that mutants lacking AP2/ERF TFs *DORNROSCHEN (DRN)* and *DORNROSCHEN-LIKE (DRNL)* cause the alteration of bilateral symmetry derived from defective PIN1-mediated IAA fluxes, similar to those described for *hd-zip III* mutant embryos [51]. Relevantly, DRN and DRNL were shown to interact in vitro with all family III members, and genetic interactions between DRN and PHV suggest that HD-ZIP III and DRN/DRNL act in the same pathway to regulate embryo symmetry via auxin distribution [51].

The role of HD-ZIP III in regulating embryonic SAM establishment also involves auxin signaling via AUXIN RESPONSE FACTOR 2 (ARF2) [52]. Mutants lacking *ARGONAUTE10/PIHNEAD/ZWILLE (AGO10/PNH/ZLL)* are unable to establish and maintain embryonic SAM because the enhanced activity of miR165/6 reduces the levels of *HD-ZIP III* [52,53]. On the other hand, *ago10* embryos display an increased expression of *ARF2* in the SAM, along with increased auxin signaling [52]. The expression of a gain-of-function version of REV restores *ARF2* expression and SAM establishment in *ago10* [52], indicating that HD-ZIP III TFs maintain embryonic SAM by tuning down auxin signaling. Similar conclusions also came from the study of *wox1235* quadruple mutants, which display defects in the establishment of embryonic SAM because of high auxin activity [54]. HD-ZIP III TFs were shown to act as effectors of the WOX2 module in tuning down auxin while promoting CK signaling to maintain the stemness of the central apical cell and preventing their differentiation [54] (Figure 1). Together, these data show that HD-ZIP III TFs are master regulators of auxin responses during embryogenesis.

Family II HD-ZIPs also play a role in embryo development by regulating bilateral symmetry, SAM determination, and root pole geometry. Plants lacking *HOMEOBOX ARABIDOPSIS THALIANA 3 (HAT3)* and *ARABIDOPSIS THALIANA HOMEOBOX 4 (ATHB4)*, two factors expressed early during embryo development, display defects in bilateral symmetry and cotyledon formation as a result of defective auxin distribution mediated by PIN1 [55]. In *hat3 athb4* mutants, and more so in the triple *hat3 athb4 athb2*, PIN1 fails to correctly polarize in the sites of incipient primordia, and its expression is further shut down later on during cotyledon development, leading to open vascular networks [55,56]. In *hat3 athb4* embryos, transient alterations in the auxin levels in the hypophysis/suspensor cells were also observed and linked to defective patterning of the root pole [57]. Interestingly, *hat3 athb4 athb2* phenotypes resemble those of multiple HD-ZIP III loss-of-function mutants, and it was shown that the introgression of either *rev* or *phb* mutations in a *hat3 athb4* background enhances the severity of bilateral symmetry and SAM specification [55,56]. These data suggest that family II and III HD-ZIPs might cooperate to regulate auxin fluxes and apical embryo development [56] (Figure 1). Relevantly, the EAR motif in the N-ter domain of some HD-ZIP II TFs is required for the interaction with TPL [58,59], leaving an open question about a possible interplay among HD-ZIP II/HD-ZIP III and TPL in regulating auxin responses during embryogenesis.

The association of HD-ZIP TFs with auxin-regulated pathways during embryogenesis is further reinforced by the discovery that a HD-ZIP I, namely ATHB5, binds to a sequence located within the promoter of *INDOLE ACETIC ACID 12/BODENLOS (IAA12/BDL)*, a key regulator of auxin responses and root pole specification downstream of IAA via TPL [45,60,61]. ATHB5 negatively regulates the expression of *BDL* in vitro, whereas in vivo, its embryonic expression pattern is complementary to that of BDL, suggesting that ATHB5 restricts *BDL* expression [60]. However, *athb5* mutants do not show embryonic defects [60], raising the possibility that ATHB5 acts redundantly with other HD-ZIP TFs.

## 4. Root Development and Patterning

All HD-ZIP TFs families play a key role in controlling the development and patterning of roots by altering hormonal responses at a local level. HD-ZIP III TFs are pivotal to the determination of the radial tissue organization and vascular development. In the root meristem, HD-ZIP III TFs *PHB*, *REV*, *CNA*, and *ATHB8* are expressed in the xylem precursors and associated procambial cells as a result of a complex regulatory loop [62,63] (Figure 2). *HD-ZIP III* expression is regulated by IAA, with auxin biosynthesis and transport playing a major role in determining their transcriptional output [63,64]. Interestingly, HD-ZIP III TFs have the capacity of feeding back on IAA levels, as auxin biosynthetic genes *TRIPTOPHAN AMINOTRANSFERASE OF ARABIDOPSIS 1 (TAA1)* and *YUCCA5 (YUC5)* and auxin transporters *PIN4* and *LIKE AUXIN RESISTANT 1 (LAX1)*, *LAX2*, and *LAX3* are direct targets of REV [65,66]. On the other hand, HD-ZIP III expression within the stele is also controlled post-transcriptionally by the endodermis-expressed miR165/6. Upon movement towards inner tissues via plasmodesmata, miR165/6 represses *PHB* and other *HD-ZIP III* transcripts, restricting their expression domain to xylem precursor/procambial cells [62,67]. The precise spatial regulation of HD-ZIP III expression within the stele is critical for correct xylem patterning as they act redundantly in a dose-dependent manner to correctly specify the development of the protoxylem (PX) and the metaxylem (MX) in outer and inner cell files, respectively [62,68]. In fact, the loss of *HD-ZIP III* expression causes PX differentiation in the inner stele, and conversely, ectopic PHB expression in miR-resistant *phb1-d* promotes MX fate in outer positions [62,68]. The correct specification of PX and MX depends on a delicate balance between IAA and CK action within the stele tissues, with a high IAA/CK ratio promoting PX [69,70]. HD-ZIP III TFs control auxin signaling by positively regulating the non-canonical auxin-resistant proteins *IAA20* and *IAA30*, and at the same time, their putative target *ARF5/MONOPTEROS (MP)* forms a signaling loop that regulates xylem development [71] (Figure 2). This regulatory module is likely direct as both *IAA20* and *MP* are transcriptional targets of PHB [71]. The capacity of HD-ZIP III TFs to interfere with auxin signaling is further confirmed by the finding that the ACAULIS 5/BUSHY AND DWARF2 (ACL5/BUD2) module acting downstream of HD-ZIP III TFs alters the expressions of several IAA-regulated genes, including auxin influx and efflux transporters [72], which are crucial to the patterning of the xylem axis [69,73]. The activity of HD-ZIP III in vascular patterning is also modulated by abscisic acid (ABA) via a non-cell-autonomous mechanism involving miR165/6 [74,75]. ABA promotes the expression of miR165/6 in the endodermis, which, in turn, reduces HD-ZIP III levels in the stele altering vascular development along the longitudinal and radial axes (Figure 2), a mechanism that might be needed to cope with adverse environmental conditions [74,75].

HD-ZIP III also affects CK signaling by restricting the expression of the negative regulator *ARABIDOPSIS HISTIDINE PHOSPHOTRANSFERASE 6 (AHP6)* in PX [62,71], controlling the IAA/CK ratio needed for proper vascular patterning [69,76]. It must be noted that *AHP6* itself is an auxin-inducible gene downstream of the MP-TARGET OF MONOTEROS 5 (TMO5)/LONESOME HIGHWAY (LHW) pathway [69,76,77]; thus, it is not clear whether *AHP6* expression is directly regulated by HD-ZIP III TFs as suggested by mathematical models [78], or rather indirectly via alterations of auxin sensitivity in the stele [71]. However, the interplay of HD-ZIP III with CK is way more complex and involves multiple regulatory mechanisms [79,80]. PHB was shown to directly regulate the biosynthetic gene *ISOPENTENYL TRANSFERASE 7 (IPT7)*, causing an increase in CK levels in the root apical meristem (RAM) [79]. High CK levels activate ARABIDOPSIS RESPONSE REGULATOR 1 (ARR1), which, in turn, regulates *PHB* transcriptionally and post-transcriptionally via miR165, forming a feedback loop that controls the proliferative balance along the vertical axis of the root [79]. On the other hand, HD-ZIP III TFs also regulate CK levels indirectly via Brassinosteroid (BR) signaling [80] (Figure 2). HD-ZIP III TFs were shown to regulate the expression of *CONSTITUTIVE PHOTOMORPHOGENIC DWARF (CPD)*, a key BR biosynthetic gene [81], controlling the local levels of the hormone [80]. BR, in turn, modulates the expression of the CK biosynthetic genes *IPT3*, *LONELY GUY 3 (LOG3)*, and *LOG4* to control procambial cell proliferation and thus root radial expansion [80] (Figure 2). Relevantly, *LOG4* was also found downstream the MP-TMO5/LHW module [76], suggesting that multiple inputs on CK biosynthesis might be required to coordinate vascular growth and patterning. Interestingly, it was recently shown that CK levels in the procambium promote the expression of a group of DNA binding with one finger (DOF) TFs collectively named PHLOEM EARLY DOF (PEAR), which form a gradient centered around phloem sieve elements that expands toward the inner xylem [82]. PEAR proteins promote the expression of *HD-ZIP III*, which, in turn, inhibit the expression of PEARs, forming a feedback loop that integrates hormone information to define patterns of procambial proliferation [82] (Figure 2). Other than regulating vascular differentiation and procambial proliferation, in older root tissues, HD-ZIP III TFs also control the establishment of a vascular cambium stem cell organizer via an auxin-mediated feedback loop in conjunction with MP, ARF9, and ARF17 [83].

HD-ZIP II TFs have been shown to play a role in controlling auxin homeostasis in the RAM by acting, to a various extent, on auxin biosynthesis, transport, and sensitivity [57,84,85]. At least five members of the HD-ZIP II family, namely *HAT3*, *ATHB4*, *ATHB2*, *HAT2*, and *HAT1*, are expressed in root meristematic tissues within the stele and the stem cell niche (SCN) [55,57,85], and among them, the latter three were shown to be auxin-inducible genes [57,84,86]. In the SCN, HAT3, ATHB4, and ATHB2 were shown to redundantly play a role in controlling cell division planes and in maintaining the proliferative status of columella stem cells (CSCs), ensuring a correct patterning of the root distal tissues [57] (Figure 2). The three HD-ZIP II TFs counteract the auxin-induced differentiation of columella cells (CCs) by promoting CSC proliferation downstream of a regulatory module controlled by the NAM/ATAF/CUC2 (NAC) TFs FEZ and SOMBRERO (SMB) [57]. HAT3, ATHB4, and ATHB2 exert their action by controlling the expressions of the class C auxin response factors *ARF10* and *ARF16* and of the efflux carriers *PIN3* and *PIN4*, leading to altered auxin responses in the RAM [57]. Relevantly, *PIN3* was also shown to be a direct target of HAT2 [85], strengthening the idea that the control of auxin fluxes via the transcriptional regulation of PIN efflux carriers may be a common function of HD-ZIP II TFs [87]. Indeed, the loss of *HD-ZIP II* expression also reduces RAM proliferation, similarly to what was observed in plants with reduced auxin transport [57,88].

Previous evidence also linked HD-ZIP II/auxin interplay to root vascular development and cambium proliferation [89], similarly to what was observed for HD-ZIP III TFs. Interestingly, functional interactions between members of HD-ZIP II and III TFs have been reported [56]. *ATHB2*, *HAT2*, *HAT3*, and *ATHB4* were shown to be direct targets of REV and possibly of other HD-ZIP III TFs [55,65]. Also, REV has been shown to interact with HAT3 and ATHB4 in complexes that are capable of repressing the transcription of *miR165/6*, thus regulating the expression of the entire HD-ZIP III family [90]. Despite clear indications of overlapping expression patterns in the stele being missing to date, it is likely that HD-ZIP II and HD-ZIP III members cooperate in the regulation of root radial patterning and vascular development. Several links have been established between HD-ZIP II TFs and hormones involved in root radial patterning, such as CK, ABA, and BR [91,92,93,94,95]. Among these, of particular interest is HAT1, which may regulate the balance between ABA and BR [94,95]. In its phosphorylated form, the HAT1 protein has been shown to interact with BRI1 EMS SUPPRESSOR 1 (BES1) and cooperatively inhibit the expression of BR-related genes, including *DWARF4 (DWF4)*, a key BR biosynthetic gene acting upstream of *CPD* [81,94]. When dephosphorylated, HAT1 has been shown to bind to the promoters of ABA-related genes, including *9-CIS-EPOXYCAROTENOID DIOXYGENASE 3 (NCED3)* and *ABA3*, repressing ABA biosynthesis and downstream responses [95]. HAT1 has been reported to be phosphorylated by the BR- and ABA-related kinases BRASSINOSTEROID INSENSITIVE 2 (BIN2) and SUCROSE NON-FERMENTING 1-RELATED PROTEIN KINASE 2.3 (SnRK2.3), respectively, with a contrasting effect of the post-translational modification on protein stability [94,95]; thus, further studies are required to shed light on the roles of HAT1 and other HD-ZIP II TFs in the regulation of BR/ABA levels and in root development.

HD-ZIP I factors have been specifically linked to BR-mediated root elongation [96]. In particular, *HAT7*, *ATHB23*, *ATHB13*, and *ATHB20* were identified by single-cell RNA-seq of root tissues as BR-upregulated genes within a cortex-specific regulatory network, in which at least *HAT7* is a direct target of BRASSINAZOLE RESISTANT 1 (BZR1) and BES1 [96]. This HD-ZIP I regulatory network is specifically active in the root transition zone (TZ) where physiologically high BR levels activate cell wall remodeling genes to promote cell expansion, and accordingly, the quadruple *hd-zip I* mutant displays shorter mature cortex cells [81,96]. Similarly, it was shown that ATHB12, another HD-ZIP I, promotes cortex cell expansion by regulating the expression of genes coding cell wall remodeling proteins, including *EXPANSIN A5 (EXPA5)*, *EXPA6*, and *EXPA15*, and also the BR biosynthetic gene *DWF4* [97,98], further strengthening the links among HD-ZIP I, BR, and cell wall loosening in the growth of root TZ cells (Figure 2).

Other than BR responses, ATHB23 is also involved in the regulation of lateral root primordia (LRP) formation downstream of auxin stimulus [99]. ATHB23 is induced by IAA via ARF7 and modulates auxin sensitivity in LRP by regulating the influx carrier *LAX3* in cooperation with LOB DOMAIN (LBD) TFs [99]. Also, the auxin-controlled ATHB23/LAX3 regulatory node has been shown to play a role in CC starch granule turnover [100]. Similar interactions between auxin transport and HD-ZIP I were observed for ATHB40/ATHB53 [101]. ATHB40 is a negative regulator of root growth, controlling RAM proliferation as well as cell elongation in the TZ [101]. ATHB40 acts downstream of auxin and ATHB53—an IAA-induced, CK-repressed HD-ZIP I [102]—to regulate auxin transporters *PIN2*, *LAX2*, and *LAX3*, with the latter being a direct target [101]. Relevantly, *PIN2* was also shown to be a direct target of ATHB52 downstream of ethylene (ET) signaling mediated by ETHYLENE INSENSITIVE 3 (EIN3) [103]. ATHB52 also controls the expression of AGC kinases *WAVY ROOT GROWTH 1* (*WAG1*) and *WAG2*, which regulate the polarity of PIN transporters in root tissues, providing a plausible mechanism linking ET signaling to IAA-mediated growth responses via HD-ZIP TFs [103].

Very few links between hormones and HD-ZIP IV in the context of root development and patterning have been uncovered to date, most of which relate to GLABRA2 (GL2). GL2 is master regulator of root epidermal patterning, which defines the fate of non-root hair cells, or atrichoblasts, by repressing hair cell (trichoblast)-specific pathways [104,105,106]. BR modulates *GL2* expression and, as a result, affects epidermal cell fate by controlling the activity of BIN2 kinase [107,108]. When phosphorylated, BIN2 positively regulates *GL2* expression via the WEREWOLF/GL3/TRANSPARENT TESTA GLABRA 1 (TTG1) transcriptional complex, establishing atrichoblast fate; in the presence of BR, unphosphorylated BIN2 inhibits *GL2* expression via TTG1 phosphorylation, promoting root hair formation [108]. The BIN2-dependent control of GL2 signaling also requires the presence of an O-glycosylated ARABINOGALACTAN PROTEIN 21 (AGP21) peptide, which is regulated by BR via BZR1 [109]. Interestingly, GL2 activity is also modulated by ET through a mechanism involving the WER/GL3/TTG1 transcriptional complex [110]. In non-hair cells, ET signals through EIN3, which, while bound on the promoter of *GL2*, interacts with GL3, thus disrupting the activity of the WER/GL3/TTG1 complex. This results in the downregulation of *GL2* with the consequent activation of root hair-specific developmental programs in atrichoblasts [110]. It is worth mentioning that the interaction between BR and ET in root epidermal cells might not be limited to establishing cell fate but may also affect cell elongation [111]. Another HD-ZIP IV with an established role in root development is HDG11 [112]. HDG11 was shown to promote jasmonic acid (JA) biosynthesis by directly regulating the expression of the biosynthetic genes *ALLENE OXIDE SYNTHASE (AOS)*, *ALLENE OXYDASE CYCLASE 3 (AOC3)*, *12-OXOPHYTODIENOATE REDUCTASE 3 (OPR3)*, and *OPC-8:0 COA LIGASE 1 (OPCL1)*. The HDG11-mediated increase in JA results in an increase in IAA signaling, affecting LRP development and root architecture [112]. Interestingly, HDG11 was also shown to promote root elongation by positively regulating genes encoding cell wall remodeling proteins [113]. It is not clear whether the cell wall loosening promoted by HDG11 is mediated by hormone signaling. However, it is worth mentioning that HDG11 was shown to interact with BRASSINAZOLE INSENSITIVE LONG HYPOCOTYL 9 (BIL9), a novel component of BR signaling involved in growth responses, although this interaction seemed to play a role in drought conditions [114].

ATML1 and PDF2 also provide an interesting link between cell expansion and hormone action in early root growth responses. It was shown that ATML1 and PDF2 are able to interact with the negative GA regulators GA INSENSITIVE (GAI), REPRESSOR OF GA1-3 (RGA), and RGA-LIKE (RGL)—collectively known as DELLA proteins—preventing the HD-ZIP IV pair to activate the transcription of growth-promoting, epidermal-specific genes [115]. The GA-mediated degradation of DELLAs releases this inhibition and allows ATML1 and PDF2 to jointly induce the transcription of their targets, including cell wall loosening genes (Figure 2), to boost growth and promote root emergence from the seed coat [115].

## 5. SAM Maintenance and Organogenesis

Several lines of evidence link HD-ZIP TFs to hormone action in the regulation of SAM activity and organ formation. Members of the HD-ZIP III family were shown to control SAM activity, radial patterning, and organogenesis to various extents [40,48,49,116,117,118]. SAM organogenesis relies on auxin accumulation in the peripheral zone (PZ) to promote organ initiation and subsequent primordium emergence [119,120]. PIN1-mediated transport promotes IAA accumulation at the PZ, causing the activation of ARF-dependent and -independent signaling pathways [32,121,122,123]. The position of auxin maxima is determined by REV, which, in conjunction with KAN1, restricts cell clusters displaying high expression and convergent patterns of PIN1 polarity within a narrow domain of the PZ [124]. This narrow domain corresponds to a region devoid of both REV and KAN1, indicating that the two TFs inhibit organ formation by controlling PIN1 expression [50,124] (Figure 3). Interestingly, both REV and KAN1 expression domains are determined by IAA; REV expression in the central zone (CZ)/inner PZ reflects high auxin levels across these regions, while KAN1 expression in the outer PZ is determined by low auxin levels at the SAM periphery and by auxin depletion at the SAM/organ boundary [124]. This peculiar organization of REV and KAN1 expression domains is further maintained throughout the emergence and serves to prepattern the dorso-ventral polarity of lateral organs [124] (Figure 3). REV/KAN1 expression patterns also determine specific hormonal domains in the SAM. For instance, transcriptomic analyses of REV-expressing domains within the SAM identified an increase in the expression of CK biosynthetic genes *LOG3* and *LOG7*, paired with a concurrent downregulation of genes involved in IAA synthesis, response, and transport [125], suggesting that REV, and probably other family III TFs, might be involved in controlling the CK/IAA ratio in the SAM (Figure 3). However, it must be pointed out that several fast IAA-responding genes were initially induced and then repressed by REV at a later time, suggesting a more complex regulation of hormone balance at the SAM [125]. The idea that HD-ZIP III TFs play a role in regulating CK/IAA balance in the SAM is also supported by studies on shoot regeneration from *calli* [64,126,127]. It was shown that PHB, REV, and PHV can interact with CK-responsive TF B-Type ARR1 and ARR2 in vitro [126]. The HD-ZIP III/B-type ARR interaction was shown to induce *WUSCHEL (WUS)* expression downstream of CK in *calli*, thus enhancing shoot regeneration, as well as in young seedlings [126]. It must be noticed that the HD-ZIP III-mediated regulation of CK levels and *WUS* could be more complex as PHB, PHV, and CNA were shown to restrict rather than promote *WUS* expression in different developmental contexts [43,128].

HD-ZIP II TFs are also known to play a role in SAM activity and organogenesis as it was shown that *hat3 athb4 athb2* mutant seedlings, in most cases, do not form viable SAMs—failing to express *WUS* and *CLAVATA3 (CLV3)*—and do not produce lateral organs [55]. However, a clear link for these TFs with hormone action in SAM activity/morphogenesis is missing to date. The three family II TFs are expressed in the SAM under the control of REV [55,65,90]; thus, it is possible that HAT3, ATHB4, and ATHB2 cooperate with HD-ZIP III to regulate the meristematic IAA/CK ratio [56]. Also, they may play a role in regulating auxin transport and sensitivity in the SAM, as observed in other tissues [55,56,57]. Interestingly, yeast one-hybrid experiments displayed that ATHB4 can bind the promoter of *ANT* [129], an AP2/ERF TF involved in organ initiation downstream of MP [32], in vitro, further suggesting that HD-ZIP II TFs contribute to auxin-mediated organogenesis.

There are very few indications with regard to a potential role of HD-ZIP I in regulating hormone-related pathways at the SAM or during primordium initiation. Four auxin-inducible HD-ZIP I members, namely *ATHB5*, *ATHB12*, *ATHB23*, and *ATHB31*, were found to be repressed by REV and excluded by REV-specific domains in the SAM, along with other genes involved in auxin homeostasis [125], suggesting that even HD-ZIP I TFs may play a role in auxin-mediated shoot morphogenesis. As previously mentioned, ATHB5 regulates the expression of *BDL* [60], which is known to regulate lateral organ initiation by inhibiting MP function [32].

HD-ZIP IV may regulate SAM activity and organogenesis in a non-cell-autonomous manner [22]. ATML1 and PDF2 have been shown to regulate the expression of *3-KETOACYL COA SYNTHASE (KCS) 20* and *KCS10/FIDDLEHEAD (FDH)*, encoding key enzymes in the synthesis of very long-chain fatty acids (VLCFAs) [115]. Other VCLFA biosynthetic genes present L1-box sequences in their promoters [130], suggesting that the entire pathway may be under HD-ZIP IV-mediated regulation. Epidermis-synthesized VLCFAs were shown to regulate cell proliferation in inner tissues—including SAM—by reducing CK biosynthesis in the vasculature [130]. Moreover, VCLFAs also control plasma membrane integrity, thus interfering with PIN1 polarity and, as a result, auxin distribution and organogenesis [131,132]. It is worth mentioning that VLCFA-containing ceramides were shown to stabilize ATML1 expression in the outer layer via its START domain, thus acting as a positional signal that sustains epidermal identity [37]. Taken together, these results suggest that HD-ZIP IV, through VCLFAs, could at once determine the epidermal cell fate and control the inner IAA/CK levels, which is in line with the idea that the epidermis drives plant growth by regulating the proliferative activity of the underlying tissues [133,134] (Figure 3). It is relevant to point out that other HD-ZIP IV members might play a more direct role in SAM activity and organogenesis. For instance, HDG1—which, together with paralogs HDG11 and HDG12, is expressed in the SAM—was shown to regulate the expression of genes involved in IAA transport, CK signaling, and GA biosynthesis along with genes involved in SAM proliferation and organogenesis [29].

## 6. Leaf Development

Leaves originate from the sides of the SAM where primordia are formed by founder initial cells. After leaf blade initiation and polarity establishment, the primordium begins to expand. Initially, cells proliferate throughout the entire leaf primordium, but with time, a transition to cell expansion, marking differentiation, occurs. The tight control of these two processes determines the final leaf size and shape.

Four out of five family III TFs, namely REV, PHV, PHB, and CNA, are considered major determinants of leaf polarity as they act by specifying adaxial cell identity in opposition with abaxial-specific KAN TFs [40,48,49,124]. As mentioned above, adaxial/abaxial identities are established at organ inception through an auxin-regulated mechanism, with REV and KAN being expressed in concentric domains separated by a narrow PIN1-expressing boundary, determining at once the site of initiation and the future dorso-ventral polarity of the primordium [124,135]. Adaxial and abaxial identities are further maintained and reinforced by mechanisms involving small RNAs: HD-ZIP III transcripts are restricted to leaf adaxial tissues by *miR165/6*, which targets HD-ZIP III mRNAs in the abaxial domain [40,136]; adaxially expressed *trans-acting siRNA 3 (TAS3)* restricts the expression of auxin response factors, including ARF3/ETTIN (ETT)—an interactor of KAN—to the abaxial domain [137,138,139]. A transcriptomic analysis at the whole seedling level highlighted that the antagonistic regulation between REV and KAN mainly occurs through opposing the regulation of downstream targets [65,66,140]. The targets Oppositely regulated by REV and KAN (ORK) include genes or genetic pathways implicated in IAA biosynthesis (*TAA1* and *YUC5*), transport (*LAX2* and *LAX3*, *WAG1*, and *NAKED PINS IN YUCCA 1*), and response (ETT), suggesting that adaxial/abaxial patterning requires a precise control of auxin homeostasis in leaf primordia [65,66,135,140]. Other than IAA, genes involved in BR, CK, and ABA pathways were also identified as differentially regulated by the REV/KAN module [140]. Genes involved in ABA perception and signaling, *PYRABACTIN RESISTANCE 1-LIKE 6 (PYL6)* and *CBL-INTERACTING SER/THR PROTEIN KINASE 12 (CIPK12)*, were identified as ORKs [140], while PHB was shown to directly regulate the expression of *ABA INSENSITIVE4 (ABI4)*, a regulator of ABA signaling, and *β-glucosidase 1 (BG1)*, which releases active ABA by hydrolyzing inactive ABA–glucose conjugates [141]. Another ORK, the family II member *HAT22*, also known as *ABA-INSENSITIVE GROWTH 1 (ABIG 1)*, was shown to be induced by ABA to regulate shoot growth in drought conditions and to regulate leaf polarity downstream of the REV/KAN module [142,143]. Interestingly, HAT22 was also identified as BRASSINOSTEROID-RELATED HOMEOBOX 3 (BHB3), a negative regulator of the BR-inducible genes *BAS1* and *SAUR-AC1* [144], supporting the hypothesis that HAT22 could pattern leaf dorsoventrality by regulating BR-mediated responses. Other family II genes, namely *HAT1*, *HAT2*, *HAT3*, *ATHB4*, and *ATHB2*, were also identified as ORKs [55,65,66,140,143]. While some HD-ZIP II TFs might contribute to the regulation of IAA transport and downstream transcriptional responses, their role in the specification of leaf dorsoventral polarity is likely related to the regulation of *miR165/6* transcription [55,90]. HAT3 and ATHB4 were shown to physically interact with REV and bind to the promoter of *miR165/6* to downregulate its expression in the adaxial domain of the leaf [90]. These data suggest that HD-ZIP II and III function together as master regulators of dorsoventral polarity and may play similar roles in auxin-mediated developmental processes [56] (Figure 3).

Leaf dorsoventral polarity is mirrored by features that characterize the adaxial and the abaxial surfaces, such as the presence of trichomes in the upper epidermis. HD-ZIP II and III TFs may also cooperate in the specification of the adaxial epidermal features, as suggested by the finding that *C2H2 zing finger proteins (ZFPs)* are targets of REV and HAT1 [140,145]. ZFP8, identified as an ORK [140], acts as a positive factor in trichome initiation downstream of GA, in a pathway that includes ZFP6 [146,147]. Recently, HAT1 was shown to negatively regulate GA-mediated trichome initiation by directly repressing the expression of *ZFP6*. Moreover, GA promotes HAT1 activity which, in turn, represses GA biosynthesis and signaling in a negative feedback loop [145]. Considering that *HAT1* is itself an ORK [140], it is possible that interactions between HD-ZIP II and III determine the specification of adaxial epidermal structures.

It is worth mentioning that ATHB8, the fifth member of family III, has a distinct function in leaf development with respect to the other HD-ZIP III [49]. ATHB8 plays a role in promoting leaf vein formation downstream of auxin [148,149,150]. During leaf development, *ATHB8* expression is activated in a single file of ground cells that will later elongate to become procambial cells, i.e., vascular precursors. The specific *ATHB8* expression in narrow pre-procambial domains is regulated by auxin via MP, which recognizes and binds to low-affinity sites in the *ATHB8* promoter, at its peak expression levels [148,149]. *ATHB8* expression is excluded outside of pre-procambial cells by an AUX/IAA-mediated inhibition of MP activity, conferring a striped pattern to ATHB8 expression and vascular development [149].

Other than controlling leaf dorsoventrality together with the HD-ZIP III family, HD-ZIP II TFs have been shown to play a role in the control of leaf cell proliferation [18] (Figure 4). Plants overexpressing *ATHB2*, *HAT1*, or *HAT2* display larger subepidermal cells compared with the wild type, suggesting that these TFs negatively regulate leaf cell proliferation [18]. Indeed, in simulated shade conditions, the concurrent action of ATHB2 and ATHB4 promotes an early differentiation of mesophyll cells, suggesting that HD-ZIP II controls the balance between cell proliferation and differentiation during leaf expansion [151]. It is possible that ATHB2 and ATHB4 control the switch between proliferation and differentiation by interfering with IAA signaling, as recently observed in roots [57]. Remarkably, a recent work identified a regulatory network including HD-ZIP II, TEOSINTE BRANCHED1/CYCLOIDEA/PCF 4 (TCP4), and auxin that controls leaf expansion and cell proliferation [152]. TCP4 promotes cell differentiation in the expanding leaf blade via an auxin-mediated upregulation of *HAT1* and *HAT2* [152]. The HD-ZIP II pair, in turn, acts by restricting cell number and leaf size. TCP4 also promotes cell differentiation by directly activating *HAT2* in an auxin-independent manner [152] (Figure 4). Interestingly, other TCP/HD-ZIP regulatory networks were shown to regulate leaf growth. TCP13, TCP5, and TCP17 inhibit leaf growth by reducing cell expansion through a mechanism involving members of families I and II of HD-ZIPs [153]. TCP13, TCP5, and TCP17 were all shown to physically interact with the C-terminal domain of ATHB2, whereas TCP13 also interacts with HAT3 and ATHB4 [153]. TCP13 downregulates—probably via the EAR repression domain of ATHB2—the expression of the HD-ZIP I TF *ATHB12* [153]. ATHB12 positively regulates cell expansion by controlling the expression of cell wall-related genes and BR biosynthesis via *DWF4* [97] (Figure 4). Relevantly, ATHB12 also acts downstream of ABA and is capable of targeting GA biosynthesis via *GA20-OXIDASE* [154], suggesting that TCP/HD-ZIP networks might represent key hubs that integrate multiple hormonal inputs to regulate leaf expansion.

HD-ZIP I TFs also control the leaf shape by restricting cell growth at margins [155]. For instance, ATHB1 controls the expression of the NAC TF CUC2 in the leaf margins by negatively regulating the expressions of *miR164A*, *B*, and *C* [156]. *miR164* is responsible for the post-transcriptional regulation of CUC2, which, in turn, controls auxin accumulation via PIN1, promoting the formation of leaf serrations [156,157]. However, it remains to be established how ATHB1—a transcriptional activator—represses *miR164* upon binding to its promoter. It is worth pointing out that *miR164* expression is also positively regulated by TCP3 [156], further strengthening the role of TCP/HD-ZIP networks in controlling leaf development (Figure 4).

While multiple HD-ZIP IV TFs play a prominent role in controlling the specification and differentiation of leaf epidermal structures, including trichomes and stomata [158], no clear intersections with hormonal pathways have been uncovered so far. However, interesting clues might come from stomata development. HDG2 was shown to promote stomata differentiation in internal leaf tissues, turning mesophyll cells into guard cells [159]. Several hormones, including IAA, BR, and ABA have been shown to regulate stomata formation and patterning [160,161,162], so it is possible that the HD-ZIP IV-mediated activation of stomata development might involve signaling via these hormonal cues.

## 7. Flower and Inflorescence Development

The development of flowers from the flank of the SAM is controlled by an intricate series of transcriptional, hormonal, and mechanical cues [163,164]. There is evidence that HD-ZIP TFs also intervene in the regulation of hormone stimuli at this developmental stage.

HD-ZIP II factors were shown to regulate auxin-mediated organ polarity switch during gynoecium development. Gynoecium morphogenesis starts as a fusion of two carpels with a bilateral symmetry, which subsequently gives rise to a completely radial style at the top end. This process is regulated by consecutive shifts in auxin distribution, which precede and guide the transition in symmetry [165]. Bilateral symmetry is associated with IAA accumulation in two lateral foci. At the beginning of the transition, two new medial foci appear, forming a four-foci structure, which subsequently evolves in a continuous, ring-formed, auxin signaling maximum [165]. A network of basic helix-loop-helix (bHLH) TFs control auxin accumulation during symmetry transition; the SPATULA/INDEHISCENT (SPT/IND) module regulates the switch from the two-foci stage to the four-foci stage [165], whereas the SPT/HECATE (HEC) module controls the progression from the four-foci maximum to the ring-shaped auxin maximum [166]. HAT3 and ATHB4 were shown to regulate auxin distribution in the gynoecium downstream of the SPT/HEC module [167]. *HAT3* and *ATHB4*, which were previously reported as direct positive targets of SPT and HECs [12,168], coordinate the radialization of the gynoecium apex from the adaxial tissue by regulating polar auxin transport via PIN1. The *hat3 athb4* double mutants show strongly reduced PIN1 expression and are blocked in the four-foci stage of the auxin maxima, resulting in a severe split gynoecium phenotype similar to that of *spt* [12,167]. The *hat3 athb4* double mutants are hypersensitive to cytokinin application, suggesting that the HD-ZIP II also controls cytokinin signaling downstream of the SPT/HEC module [167]. In other words, this suggests that HD-ZIP II/bHLH TF networks regulate gynoecium morphogenesis by controlling the IAA/CK balance. Intriguingly, HAT1—also known as JAIBA—was shown to play a role in reproductive development [169]. In a recent protein–protein interaction analysis of TFs involved in gynoecium development, HAT1 has been found to interact with the B-type ARR14, a positive regulator of CK responses [170], suggesting that—similarly to HAT3 and ATHB4—HAT1 could control gynoecium development via CK signaling. Interestingly, the same interactome analysis revealed that ARR14 also interacts with PHV, which, in turn, can also interact with ARF19 [170]. Also, REV was shown to act synergistically with ANT to control auxin homeostasis during gynoecium development [171]. While these data suggest that HD-ZIP III proteins might modulate in IAA and CK during flower development, no other pieces of evidence can confirm this hypothesis. However, family III members are required throughout the entire flower development process—from organ inception to ovule development—as well as for the specification of axillary meristems [172,173,174], so it is likely that further links with hormone pathways are going to be revealed in the near future.

HD-ZIP IV TFs were shown to be expressed in the flower epidermis and to play a role in flower development by altering the expressions of the stamen and petal identity gene *AP3* in a non-cell-autonomous fashion [29,175]. Interestingly, while studying the role of ATML1 in sepal giant cell specification, a recent preprint showed that ATML1 regulates the synthesis of VLCFA/LCFAs to promote endoreduplication [176], leaving an open possibility that, even in flowers, HD-ZIP IV-induced lipidic molecules might promote non-cell-autonomous growth by altering the hormonal balance in inner tissues, as discussed above.

HD-ZIP I TFs have been linked to the development of inflorescence. For instance, ATHB12, whose expression is regulated by ABA, was found to negatively regulate the growth of inflorescence stem, repressing *GA 20-OXIDASE* expression [154]. Interestingly, three family I members, namely ATHB21, ATHB40, and ATHB53, which are expressed in axillary buds downstream of the TCP TF BRANCHED 1 (BRC1), promote the expression of the ABA biosynthetic gene *NCED3*, leading to ABA-induced bud dormancy [177]. Recently, ATHB40 has been shown to be part of an autoregulative loop with GA [178], further linking HD-ZIP I to ABA/GA interactions. It is worth pointing out that *ATHB51*, also known as *LATE MERISTEM IDENTITY 1 (LMI1)*, is a positive direct target of LEAFY (LFY), thus acting in the early phases of flower development downstream of IAA, and it promotes flower identity through a regulatory loop together with LFY [179].

## 8. Concluding Remarks and Perspectives

Since their first identification in the early 1990s, HD-ZIPs have been the subject of intensive research efforts and emerged as essential regulators of plant development. From the body of work we discussed, auxin and cytokinin emerge as preferential targets of HD-ZIP action in different tissues. IAA and CK have been known for a long time to have an antagonistic effect on plant development by controlling the balance between proliferation and differentiation [180]. Interestingly, HD-ZIPs also target other antagonistic hormone pairs such as ABA/GA or ABA/BR [180], suggesting that they might be shifting hormone balances to orchestrate the fine shaping of the plant architecture. In the same vein, a number of studies have shown that HD-ZIPs target genes coding cell wall remodeling proteins either directly or by regulating growth-promoting hormones like IAA, BR, or GA. These data suggest that HD-ZIPs regulate plant shape by modifying the mechanical properties of single cells and tissues.

It is worth pointing out that a compelling amount of evidence also implicated HD-ZIP in the regulation of plant responses and adaptations to altered environmental conditions, including light, temperature, salinity, and stress [75,114,181,182,183]. This clearly indicates that HD-ZIPs can shape plant architectures in many different forms and under various external stimuli. This plasticity might be explained by the peculiar ability of HD-ZIP TFs to integrate into larger regulatory networks and control the action of different hormones at many levels, as discussed above. Thus, HD-ZIPs may represent the molecular hub that integrates internal and external cues to regulate plant development and synchronize growth to fluctuating environmental conditions. In view of this, we anticipate that in the coming years, research on HD-ZIP TFs will be key to the development of new translational applications to increase crop productivity and resilience and help agriculture face the current challenges of climate change and food security.

## Figures and Tables

**Figure 1 ijms-25-05657-f001:**
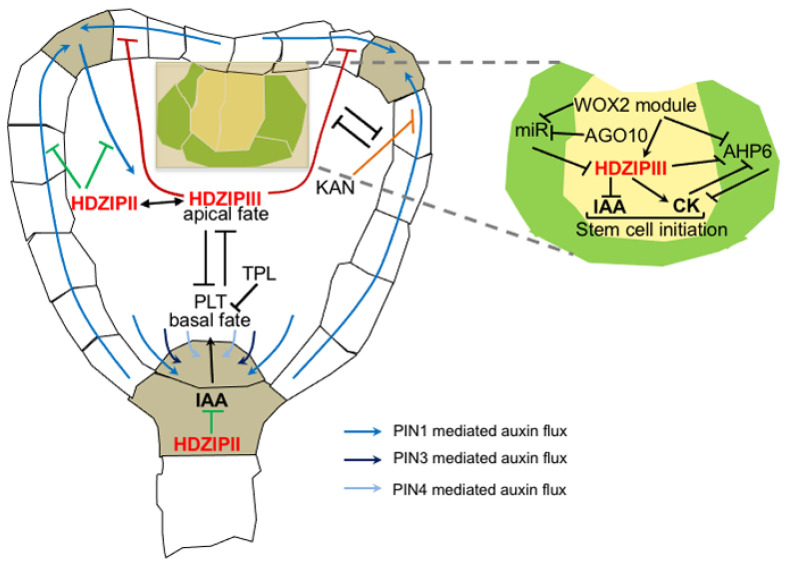
Interactions between HD-ZIP II and III in controlling auxin-mediated embryo development. HD-ZIP III determines the apical fate by antagonizing the function of the auxin-regulated, root-pole-specific PLTs (black lines); the bilateral symmetry controlling PIN1-mediated auxin fluxes at the site of incipient cotyledons (dark red lines) in opposition to KAN factors (orange line), but possibly in combination with HD-ZIP II factors (green line); and the establishment of SAM by balancing IAA and CK activity (inset). At the embryonic root pole, HD-ZIP II factors regulate auxin levels to control stem cell patterning (green lines).

**Figure 2 ijms-25-05657-f002:**
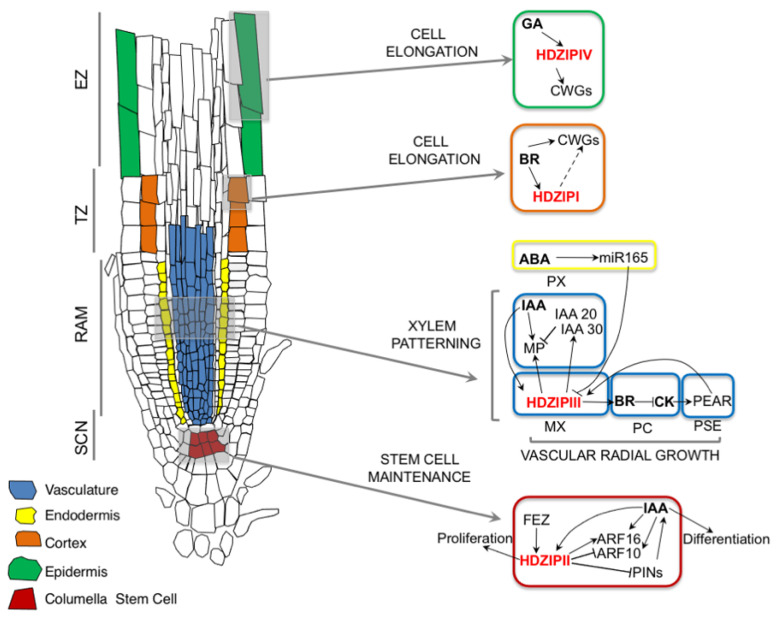
Schematic representation of HD-ZIP/hormone networks in root development and patterning. HD-ZIP I and IV promote cell elongation in TZ downstream of BR and in elongation zone (EZ) downstream of GA, respectively. Both HD-ZIP I and HD-ZIP IV target cell wall remodeling (CWR) genes to promote cell elongation. HD-ZIP III members regulate radial root patterning and procambial cell (PC) proliferation by interfering with multiple hormones, including IAA, CK, BR and ABA. HD-ZIP III interaction with CK also regulates cell differentiation at TZ. PSE, phloem sieve element. HD-ZIPII members regulate stem cell fate by counteracting IAA-mediated CSC differentiation by regulating auxin transport via PIN and auxin signaling via Class C ARFs.

**Figure 3 ijms-25-05657-f003:**
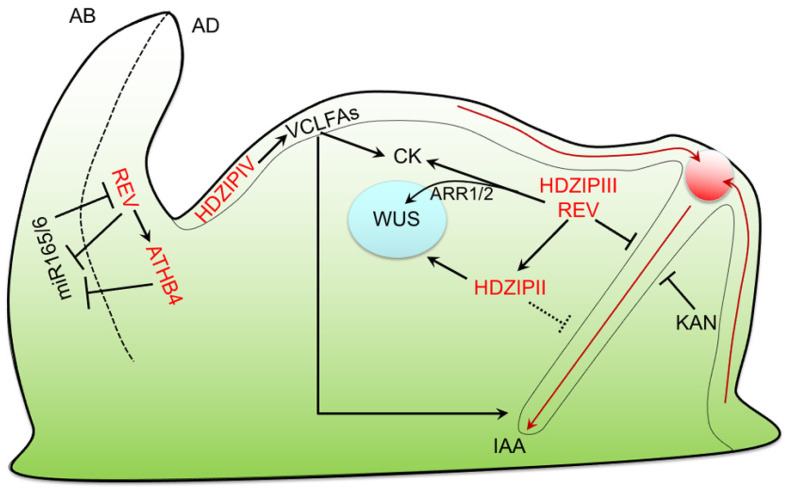
Schematic representation of HD-ZIP/hormone networks in SAM. REV and KAN restrict PIN1-mediated transport (red arrows) to narrow region of cells that will accumulate auxin (red dot) and give rise to primordium. REV and other HD-ZIP III TFs also interfere with CK synthesis and transport, affecting WUS expression and SAM identity. HD-ZIP II members are regulated by REV and may contribute to regulation of auxin transport via PIN1 and are required for WUS expression. Members of HD-ZIP II and III families also regulate primordium dorso-ventral polarity by repressing miR165/6. HD-ZIP IV regulates VCLFA biosynthesis in L1 layer. VCLFAs, by concurrently acting on CK and PIN1-mediated transport, might regulate proliferation in inner tissues.

**Figure 4 ijms-25-05657-f004:**
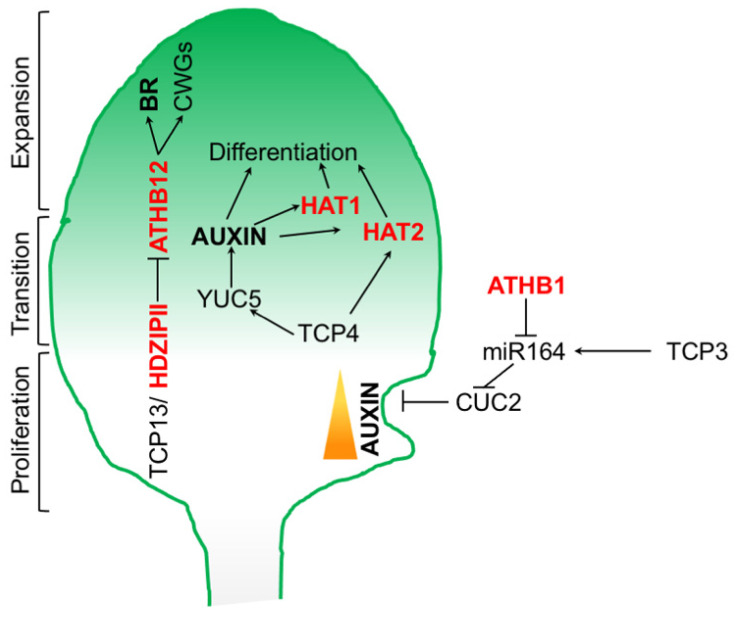
HD-ZIP/TCP networks control the leaf size and shape. The HD-ZIP II factors HAT1 and HAT2 act downstream of TCP4 and IAA to promote leaf cell differentiation. *HAT2* is also a direct target of TCP4. TCP13, and paralogs TCP5 and TCP17, interact with HD-ZIP II TFs to control leaf expansion. TCP13, likely together with ATHB2, negatively controls the leaf size by repressing ATHB12 which, in turn, promotes cell expansion by acting on BR biosynthesis and cell wall genes (CWGs). ATHB1 promotes leaf serration by controlling CUC2 expression via *miR164*. CUC2 regulates leaf serration by modulating auxin accumulation at the leaf margin. *miR164* expression is positively regulated by TCP3.

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
