# Peer review of "The Ins and Outs of Homeodomain-Leucine Zipper/Hormone Networks in the Regulation of Plant Development"

_ijms, 2024, doi:10.3390/ijms25115657_

Round 1
Reviewer 1 Report
Comments and Suggestions for Authors
I am writing to provide my evaluation of the review paper submitted to consider for publish in Int. J. Mol. Sci.
My overall assessment of the document, entitled “The ins and outs of Homeodomain-Leucine Zipper-hormone networks in the regulation of plant development” is that the quality and quantity of compiled work is indeed sufficient for publication.
Given that the manuscript is well-written and thorough, I have relatively few comments, which are listed below.
- I suggest that the authors remove auxin; and cytokinin from the keywords.
- In the abstract section, the author spent too much space introducing the purpose and significance, without spending more space on the HD-ZIP transcription factor.
- I suggest that the authors change “3. Embryogenesis” to “3. Role of HD-ZIP TFs in embryo development”.
- I suggest that the authors change “4. Root development and patterning” to “4. Role of HD-ZIP TFs in root development”.
- I suggest that the authors change “5. SAM maintenance and organogenesis” to “4. Role of HD-ZIP TFs in SAM maintenance and organogenesis”.
- I suggest that the authors change “6. Leaf Development” to “6. Role of HD-ZIP TFs in leaf development”.
- I suggest that the authors change “7. Flower and inflorescence development” to “7. Role of HD-ZIP TFs in flower and inflorescence development”.
- I suggest that the authors change “8. Conclusions” to “8. Conclusions and future prospects”.
These are my only (minor) comments on this paper.
Comments on the Quality of English LanguageMinor editing of English language required.
Author Response
I am writing to provide my evaluation of the review paper submitted to consider for publish in Int. J. Mol. Sci.
My overall assessment of the document, entitled “The ins and outs of Homeodomain-Leucine Zipper-hormone networks in the regulation of plant development” is that the quality and quantity of compiled work is indeed sufficient for publication.
Given that the manuscript is well-written and thorough, I have relatively few comments, which are listed below.
We thank this reviewer for their kind comment on our manuscript. Point-by-point responses are given below.
1. I suggest that the authors remove auxin; and cytokinin from the keywords.
1) We do agree with this reviewer and we have now removed auxin and cytokinin from the manuscript keywords.
2. In the abstract section, the author spent too much space introducing the purpose and significance, without spending more space on the HD-ZIP transcription factor.
2) We do understand this reviewer’s remark on the abstract. However, we do believe that increasing the emphasis on HD-ZIPs in the abstract could mislead the reader about the scope of this manuscript. This review is meant to highlight and connect all the significant interactions among HD-ZIP and hormones with a specific focus on plant development, and it is aimed at a Special Issue titled “Regulation of Transcription Factor–Hormone Networks in Plants”. Indeed, the abstract was purposefully written in a way that sets the background on the topic TF/Hormone networks with a developmental standpoint and furher narrows down the narrative on HD-ZIP TFs and thir interaction with hormonal networks in different developmental contexts, highlighting the actual scope of the review in the last sentences.
3. I suggest that the authors change “3. Embryogenesis” to “3. Role of HD-ZIP TFs in embryo development”.
3) As discussed above, the scope of this review is not on HD-ZIP TFs per se but rather on HD-ZIP/hormone networks so the suggested title “3. Role of HD-ZIP TFs in embryo development” could be equally misleading as it only emphasizes the TF component of the networks. To fit with the scope of the manuscript the correct section title should be “3. Role of HD-ZIP/hormone networks in embryo development” which may be clunky and redundant with the title of the review, more so if also applied to the other sections as suggested below.
4. I suggest that the authors change “4. Root development and patterning” to “4. Role of HD-ZIP TFs in root development”.
4) See point 3
5. I suggest that the authors change “5. SAM maintenance and organogenesis” to “4. Role of HD-ZIP TFs in SAM maintenance and organogenesis”.
5) See point 3
6. I suggest that the authors change “6. Leaf Development” to “6. Role of HD-ZIP TFs in leaf development”.
6) See point 3
7. I suggest that the authors change “7. Flower and inflorescence development” to “7. Role of HD-ZIP TFs in flower and inflorescence development”.
7) See point 3
8. I suggest that the authors change “8. Conclusions” to “8. Conclusions and future prospects”.
8) We do agree with this reviewer, and we have changed the title of section 8 accordingly. Section 8 is now titled "Concluding remarks and perspectives"
These are my only (minor) comments on this paper.
Reviewer 2 Report
Comments and Suggestions for Authors
In my opinion, the review article is really well written. The article focuses on the importance of Homeodomain-Leucine Zipper (HD-ZIP) proteins, a class of plant-specific transcriptional regulators, in the regulation of plant development and synchronization of development under different environmental conditions. The article develops theories about the role of HD-ZIP proteins as key regulators of hormones in controlling plant development, from embryogenesis to the formation of different organs. In summary, HD-ZIP proteins may represent important candidates for work toward improving plant adaptation to adverse environmental conditions, which is increasingly important given that we need to deal with climate change and adverse environmental events such as drought, excessive heat, cold, etc.
As for the references used, I consider them appropriate within the context developed in the text. The figures are relatively complex, but they present the theoretical content presented in the text relatively clearly.
In short, I consider the article important to add scientific knowledge and good for publication.
The article is a very well written review of current knowledge about HD-ZIP/hormone networks in the regulation of plant architecture and tissue growth across different developmental stages of plants.
Comments on the Quality of English LanguagePlease, correct the third figure legend to Figure 3 and check possible typo error in the sentence on lines 474-478.
Author Response
In my opinion, the review article is really well written. The article focuses on the importance of Homeodomain-Leucine Zipper (HD-ZIP) proteins, a class of plant-specific transcriptional regulators, in the regulation of plant development and synchronization of development under different environmental conditions. The article develops theories about the role of HD-ZIP proteins as key regulators of hormones in controlling plant development, from embryogenesis to the formation of different organs. In summary, HD-ZIP proteins may represent important candidates for work toward improving plant adaptation to adverse environmental conditions, which is increasingly important given that we need to deal with climate change and adverse environmental events such as drought, excessive heat, cold, etc.
As for the references used, I consider them appropriate within the context developed in the text. The figures are relatively complex, but they present the theoretical content presented in the text relatively clearly.
In short, I consider the article important to add scientific knowledge and good for publication.
The article is a very well written review of current knowledge about HD-ZIP/hormone networks in the regulation of plant architecture and tissue growth across different developmental stages of plants.
Please, correct the third figure legend to Figure 3 and check possible typo error in the sentence on lines 474-478.
We thank this reviewer their kind comments on our manuscript. We have now amended the mistakes in the legend of Figure 3 and corrected the typo in the sentence at lines 474-478.
Reviewer 3 Report
Comments and Suggestions for Authors
This review manuscript entitled "The ins and outs of Homeodomain-Leucine Zipper-hormone networks in the regulation of plant development" provides fundermental insight of homeodomain-leucine zipper/hormone networks in plants. This paper is well written, so I recommend publication of this manuscript for the IJMS.
minor:
I think it would be good for readers to enhance understanding, if there is table data for the genes or proteins involved
in stage- or tissue-specific development of HD-ZIP/hormone network in plants (lead development, SAM organogenesis, root development, etc.).
Author Response
This review manuscript entitled "The ins and outs of Homeodomain-Leucine Zipper-hormone networks in the regulation of plant development" provides fundermental insight of homeodomain-leucine zipper/hormone networks in plants. This paper is well written, so I recommend publication of this manuscript for the IJMS.
We thank this reviewer for their kind comments on our manuscript.